# Temporal trends and socioeconomic differences in acute respiratory infection hospitalisations in children: an intercountry comparison of birth cohort studies in Western Australia, England and Scotland

Hannah C Moore,[1] Nicholas de Klerk,[1] Christopher C Blyth,[1,2,3,4] Ruth Gilbert,[5] Parveen Fathima,[1] Ania Zylbersztejn,[5] Maximiliane Verfürden,[5] Pia Hardelid[5]

For numbered affiliations see end of article.

**Correspondence to**
Dr Hannah C Moore;
Hannah.Moore@telethonkids.org.au

## ABSTRACT

**Objectives** Acute respiratory infections (ARIs) are a global cause of childhood morbidity. We compared temporal trends and socioeconomic disparities for ARI hospitalisations in young children across Western Australia, England and Scotland.

**Design** Retrospective population-based cohort studies using linked birth, death and hospitalisation data.

**Setting and participants** Population birth cohorts spanning 2000–2012 (Western Australia and Scotland) and 2003–2012 (England).

**Outcome measures** ARI hospitalisations in infants (<12 months) and children (1–4 years) were identified through International Classification of Diseases, 10th edition diagnosis codes. We calculated admission rates per 1000 child-years by diagnosis and jurisdiction-specific socioeconomic deprivation and used negative binomial regression to assess temporal trends.

**Results** The overall infant ARI admission rate was 44.3/1000 child-years in Western Australia, 40.7/1000 in Scotland and 40.1/1000 in England. Equivalent rates in children aged 1–4 years were 9.0, 7.6 and 7.6. Bronchiolitis was the most common diagnosis. Compared with the least socioeconomically deprived, those most deprived had higher ARI hospitalisation risk (incidence rate ratio 3.9 (95% CI 3.5 to 4.2) for Western Australia; 1.9 (1.7 to 2.1) for England; 1.3 (1.1 to 1.4) for Scotland. ARI admissions in infants were stable in Western Australia but increased annually in England (5%) and Scotland (3%) after adjusting for non-ARI admissions, sex and deprivation.

**Conclusions** Admissions for ARI were higher in Western Australia and displayed greater socioeconomic disparities than England and Scotland, where ARI rates are increasing. Prevention programmes focusing on disadvantaged populations in all three countries are likely to translate into real improvements in the burden of ARI in children.

## BACKGROUND

Acute respiratory infections (ARIs) including bronchiolitis, pneumonia and influenza

---

### Strengths and limitations of this study

► We used population-level data from three countries to assess hospitalisation rates and changes over time for acute respiratory infections (ARIs) in children.
► Analysis protocols and diagnosis coding was standardised across each country.
► Hospitalisation rates for ARIs were described according to level of socioeconomic deprivation.
► To control for changing admission thresholds within each country, we adjusted our models for all non-ARI admissions.
► A limitation of this study is the different measures of socioeconomic deprivation available across the three countries.

---

are a major cause of hospitalisation in children worldwide, responsible for ~12 million annual episodes in children under 5 years of age.[1 2] In England, the hospital admission rate for ARI increased by 40% from 1999 to 2010 among children aged <15 years[3] and bronchiolitis was the most common reason for unplanned admissions in infants from 2010 to 2013.[4] While hospitalisations for ARI doubled from 1992 to 2000 in Western Australia,[5] they since stabilised 2000–2005.[6] Vaccination programmes including influenza, pertussis and pneumococcal disease have been implemented in North America, Europe and Australia, but the majority of ARI hospitalisations in high-income countries are now caused by non-vaccine preventable viruses including respiratory syncytial virus (RSV), parainfluenza virus and human metapneumovirus.[7]

ARI hospitalisations are more common among children from poorer socioeconomic backgrounds.[8 9] In addition to access to inadequate healthcare, risk factors for developing severe symptoms of ARIs, including prematurity, low birth weight, congenital anomalies, exposure to environmental tobacco smoke, damp and mould, and household overcrowding are all more common among children growing up in more deprived families in both high-income and low-income settings.[10 11] Understanding the impact of socioeconomic disparities on ARI hospitalisations among children (both over time and between countries) can provide an estimate of the preventable proportion of ARI. Linkage of administrative health datasets provides a platform to investigate these trends in populations over many years. Additionally, the availability of comparable hospital admission datasets with similar coding systems using International Classification of Diseases, 10th edition (ICD-10) diagnosis codes allows comparison of hospitalisation rates among children for ARI according to deprivation level.

Using record linkage resources within Western Australia, England and Scotland, we conducted a comparative analysis of the three jurisdictions to investigate the hospitalisation rates for ARI in children aged <5 years. All three jurisdictions have publicly funded healthcare with free access to primary and public hospital care. Each jurisdiction has established childhood vaccination programmes targeting ARIs. This includes diphtheria, tetanus, pertussis, *Haemophilus influenzae* type B (three dose infant schedule), pneumococcal disease (2+1 schedule) and recently, seasonal influenza. Excluding influenza, vaccination coverage at age 12 months is >90% for all three jurisdictions.[12 13] Our aim was to compare population-based hospitalisation rates by ARI diagnosis, age and level of socioeconomic deprivation and assess how ARI hospitalisation rates have changed over time.

## METHODS
### Data sources and study populations
We conducted separate population-based birth cohort studies using administrative data from Western Australia, England and Scotland. Western Australia covers the western third of Australia, an area of 2.5 million square kilometres with a population of ~2.6 million,[14] 3.6% of whom identify as being Aboriginal and/or Torres Strait Islander (herein referred to as Aboriginal).[15] Births were identified from the Midwives' Notification System and Birth Register, deaths were identified from the death register and hospitalisations were recorded in the Hospital Morbidity Database Collection that provides full coverage of all hospital separations (hereafter referred to as hospitalisations). In the absence of a unique person identifier in Australia, extracted data were probabilistically linked by the Western Australian Data Linkage Branch using a series of demographic identifiers using an established best practice protocol.[16 17] Aboriginal status was derived using a validated algorithm using Aboriginal identification

information across all available records.[18] England has a population of 53.9 million.[19] The birth cohort was established by linking hospital birth and delivery records from the Hospital Episode Statistics database.[20] Hospitalisations and deaths were identified via linkage to mortality registration data from the Office for National Statistics.[21] Data linkage in England was carried out by National Health Service (NHS) Digital, using a deterministic algorithm based on the NHS number (a unique patient identifier in the English NHS), postcode, date of birth, sex and local hospital numbers. Scotland has a population of 5.3 million.[19] The Scottish birth cohort was developed through linking data from birth registration and maternity databases.[22 23] Hospitalisations and deaths were identified via linkage to the Scottish Morbidity Record 01 and mortality records using deterministic linkage carried out by the electronic Data Research and Innovation Service based on the Community Health Index number, a unique identifier recorded on all births and subsequent encounters within the Scottish NHS.

The datasets represented 99.9% of all births in Western Australia, 97.5% in England[24] and 100% in Scotland with full coverage of inpatient and day admissions. Our study population comprised of singleton births in Western Australia and Scotland 2000–2012 and England 2003–2012. Multiple births were excluded due to a higher likelihood of linkage error. Children were followed from birth until their fifth birthday, date of death or 30 June 2013 (the end of follow-up) or (Scotland only) date of emigration, whichever occurred first.

### Outcome measures
Our outcome measure was an ARI emergency hospitalisation for children in their first 5 years of life. All inter-hospital transfers were collapsed into a single admission. We identified hospitalisations for ARI using a selection of ICD-10 diagnosis codes (ICD-10-AM for Western Australia).[25] Hospitalisation data for each jurisdiction provided a principal diagnosis code and up to 20 secondary diagnosis codes in Western Australia, 19 in England and 5 in Scotland. We identified ARI hospitalisations using the principal diagnosis code and all the available additional diagnosis codes as six diagnostic groups: whooping cough (A37), influenza (J09-J11), pneumonia (J12-J18, B59, B05.2, B37.1, B01.2), bronchitis (J20, J40), bronchiolitis (J21) and unspecified acute lower respiratory infection (ALRI) (J22). Consistent with our previous Western Australian work,[6] ARI hospitalisations within 14 days of a previous ARI hospitalisation were classified as a single infection episode. In such cases, we applied a hierarchical diagnosis algorithm[6] within the readmission set in order to code an overall principal diagnosis. This algorithm ranked diagnoses in order of disease severity: whooping cough, pneumonia, bronchiolitis, influenza, unspecified ALRI and bronchitis. Children with missing data on sex or deprivation were excluded from the analyses. Deaths due to ARI in these populations are rare and our data would be not sufficiently powered to assess mortality rates

in this cohort, especially for Western Australia and Scotland. As such we do not report ARI-related mortality rates here and focus our outcome measure on ARI-related hospitalisations.

## Exposure measures

It is known that hospitalisation rates for ARI are higher in infants aged <12 months than those aged older than 12 months. Thus, we assessed hospitalisations for ARI in infants aged <12 months and young children aged 1–4 years at time of admission. Other exposure measures of interest were sex, level of socioeconomic deprivation and admission year. In Western Australia, socioeconomic deprivation was measured through the index of relative disadvantage (IRSD), one of the four socioeconomic indexes for areas derived by the Australian Bureau of Statistics.[26] The IRSD score is derived from 17 different variables including low income, internet connection, unemployment and education.[26] Scores were grouped into collectors district, the smallest unit for population-based analyses which, on average, consist of ~200 dwellings. For England, socioeconomic deprivation was measured through the index of multiple deprivation (IMD), based on seven domains of deprivation including income, employment, education, crime, barriers to housing and living environment.[27] IMD scores are measured at lower super output area level, covering an average of 1200–1500 households. For Scotland, deprivations scores were based on the Carstairs index, based on four variables including car ownership, male unemployment, overcrowding and low occupational social class. The Carstairs index is measured at postcode sector level, which contains an average of 5000 people.[28] In all jurisdictions, socioeconomic deprivation scores were based on mother's residential address at time of her child's delivery and were grouped into four levels based on a country-level ranking with the lowest scores representing the most socioeconomically deprived.

## Statistical analysis

Consistent methodology was applied to the assembled datasets in the three jurisdictions. We calculated hospitalisation rates per 1000 child-years at risk for each diagnostic grouping of ARI (as principal diagnosis). To assess the impact of including additional diagnosis codes, we compared hospitalisation rates derived using the principal diagnosis code only with rates derived from using the principal plus all additional diagnosis codes (any diagnosis). We used any diagnosis to assess ARI rates by socioeconomic deprivation and year of admission. We present age-specific hospitalisation rates with 95% CIs and where appropriate, rates were compared using incidence rate ratios (IRRs) with 95% CIs. To assess temporal trends, we plotted annual hospitalisation rates in the two age groups for each jurisdiction by admission year for all ARIs and bronchiolitis, pneumonia and unspecified ALRIs. We also used negative binomial regression models to assess linear temporal trends in infant hospitalisations from 2001 to 2012 (Western Australia and Scotland) and 2004 to 2012

(England). Year of admission was included as a linear term in the models, and the natural logarithm of child-years at risk was included as an offset in the models. Trends over time in ARI admission rates were assumed to be statistically significant if the Wald test p value for the coefficient for the linear year term was <0.05. Models were adjusted for sex and the four-level socioeconomic indicator and we present IRRs with 95% CIs. In order to control for overall trends in hospitalisation, we also adjusted the models for the number of all non-ARI emergency admissions.[29] All data analyses were conducted within each jurisdiction in Stata V.14.0.[30]

## Patient and public involvement

A community reference group located in Western Australia was consulted during the conduct of this study. No individual patients were involved.

## RESULTS

A total of 337909 (Western Australia), 5939009 (England) and 699590 (Scotland) births were included in the study (see online supplementary table 1). There were 14480 infant hospitalisations for ARI as a principal diagnosis in Western Australia, 217985 for England and 26103 for Scotland giving overall infant hospitalisation rates of 44.3/1000 child-years for Western Australia, 40.7/1000 for Scotland and 40.1/1000 for England. In all jurisdictions, bronchiolitis had the highest hospitalisation rates accounting for 79% of ARI admissions in infants in Western Australia, 79% in England and 84% in Scotland (table 1). ARI hospitalisation rates in infants were higher in Western Australia compared with England and Scotland across all ARI diagnoses, most notably for pneumonia, where rates were 1.4–2.2 times higher compared with England and Scotland. The only exception was for unspecified ALRI where the hospitalisation rate in infants was 70% higher in England than in Scotland and Western Australia. ARI hospitalisation rates in children aged 1–4 years were 19% higher in Western Australia compared with England and Scotland (table 1). The most common ARI principal diagnosis among children aged 1–4 years was pneumonia in Western Australia (42%) and unspecified ALRI in England (54.6%) and Scotland (43.9%). Consequently, hospitalisation rates for pneumonia in Western Australian children aged 1–4 years were 1.5–1.8 times higher than England and Scotland.

When ARI hospitalisations were identified based on any diagnosis compared with principal diagnosis only, the difference in hospitalisation rates varied across diagnoses with the most notable difference for unspecified ALRI in Western Australia where rates were 1.5 (95% CI 1.4 to 1.6) times higher in infants when using any diagnosis compared with principal diagnosis only (see online supplementary table 2).

ARI hospitalisation rates were higher for children from the most socioeconomically deprived areas. The association with deprivation was greatest in Western Australia

**Table 1** Number of admissions and hospitalisation rate for ARI by diagnostic category by principal diagnosis in infants aged <1 year and children aged 1–4 years in Western Australia, England and Scotland

| Diagnosis | Western Australia | | England | | Scotland | |
|---|---|---|---|---|---|---|
| | n (%) | Rate (95% CI)* | n (%) | Rate (95% CI)* | n (%) | Rate (95% CI)* |
| **<1 year†** | | | | | | |
| Whooping cough | 220 (1.6) | 0.7 (0.6 to 0.8) | 2395 (1.1) | 0.5 (0.4 to 0.5) | 372 (1.4) | 0.6 (0.5 to 0.6) |
| Pneumonia | 1278 (9.4) | 4.1 (3.9 to 4.4) | 15 592 (7.2) | 2.9 (2.9 to 3.0) | 1245 (4.8) | 1.9 (1.8 to 2.1) |
| Bronchiolitis | 10 652 (78.7) | 34.4 (33.8 to 35.1) | 171 805 (78.8) | 32.2 (32.1 to 32.4) | 22 021 (84.4) | 34.3 (33.9 to 34.8) |
| Influenza | 407 (3.0) | 1.3 (1.2 to 1.4) | 1627 (0.7) | 0.3 (0.3 to 0.3) | 426 (1.6) | 0.7 (0.6 to 0.7) |
| Unspecified ALRI | 809 (6.0) | 2.6 (2.4 to 2.8) | 24 563 (11.3) | 4.6 (4.5 to 4.7) | 1797 (6.9) | 2.8 (2.7 to 2.9) |
| Bronchitis | 169 (1.2) | 0.5 (0.5 to 0.6) | 2003 (0.9) | 0.4 (0.4 to 0.4) | 242 (0.9) | 0.4 (0.3 to 0.4) |
| All ARI | 13 535 (100.0) | 43.7 (43.0 to 44.5) | 217 985 (100.0) | 40.1 (40.7 to 41.1) | 26 103 (100.0) | 40.7 (40.2 to 41.2) |
| **1–4 years‡** | | | | | | |
| Whooping cough | 33 (0.4) | 0.04 (0.03 to 0.06) | 95 (0.1) | 0.008 (0.007 to 0.01) | 23 (0.2) | 0.01 (0.01,0.02) |
| Pneumonia | 3031 (41.6) | 3.7 (3.6 to 3.9) | 29 741 (33.2) | 2.5 (2.5 to 2.6) | 3411 (26.9) | 2.1 (2.0 to 2.1) |
| Bronchiolitis | 1893 (26.0) | 2.3 (2.2 to 2.4) | 8283 (9.2) | 0.7 (0.7 to 0.7) | 3141 (24.7) | 1.9 (1.8 to 2.0) |
| Influenza | 366 (5.0) | 0.4 (0.4 to 0.5) | 1714 (1.9) | 0.2 (0.1 to 0.2) | 392 (3.1) | 0.2 (0.2 to 0.3) |
| Unspecified ALRI | 1767 (24.3) | 2.2 (2.1 to 2.3) | 48 910 (54.6) | 4.2 (4.1 to 4.2) | 5570 (43.9) | 3.3 (3.3 to 3.4) |
| Bronchitis | 195 (2.7) | 0.2 (0.2 to 0.3) | 859 (1.0) | 0.1 (0.1 to 0.1) | 161 (1.3) | 0.1 (0.1 to 0.1) |
| All ALRI | 7285 (100.0) | 9.0 (8.8 to 9.2) | 89 602 (100.0) | 7.6 (7.6 to 7.7) | 12 698 (100.0) | 7.6 (7.5 to 7.8) |

*Rate is per 1000/child-years.
†2001–2012 for Western Australia and Scotland; 2004–2012 for England.
‡2005–2012 for Western Australia and Scotland; 2008–2012 for England.
ALRI, acute lower respiratory infection; ARI, acute respiratory infection.

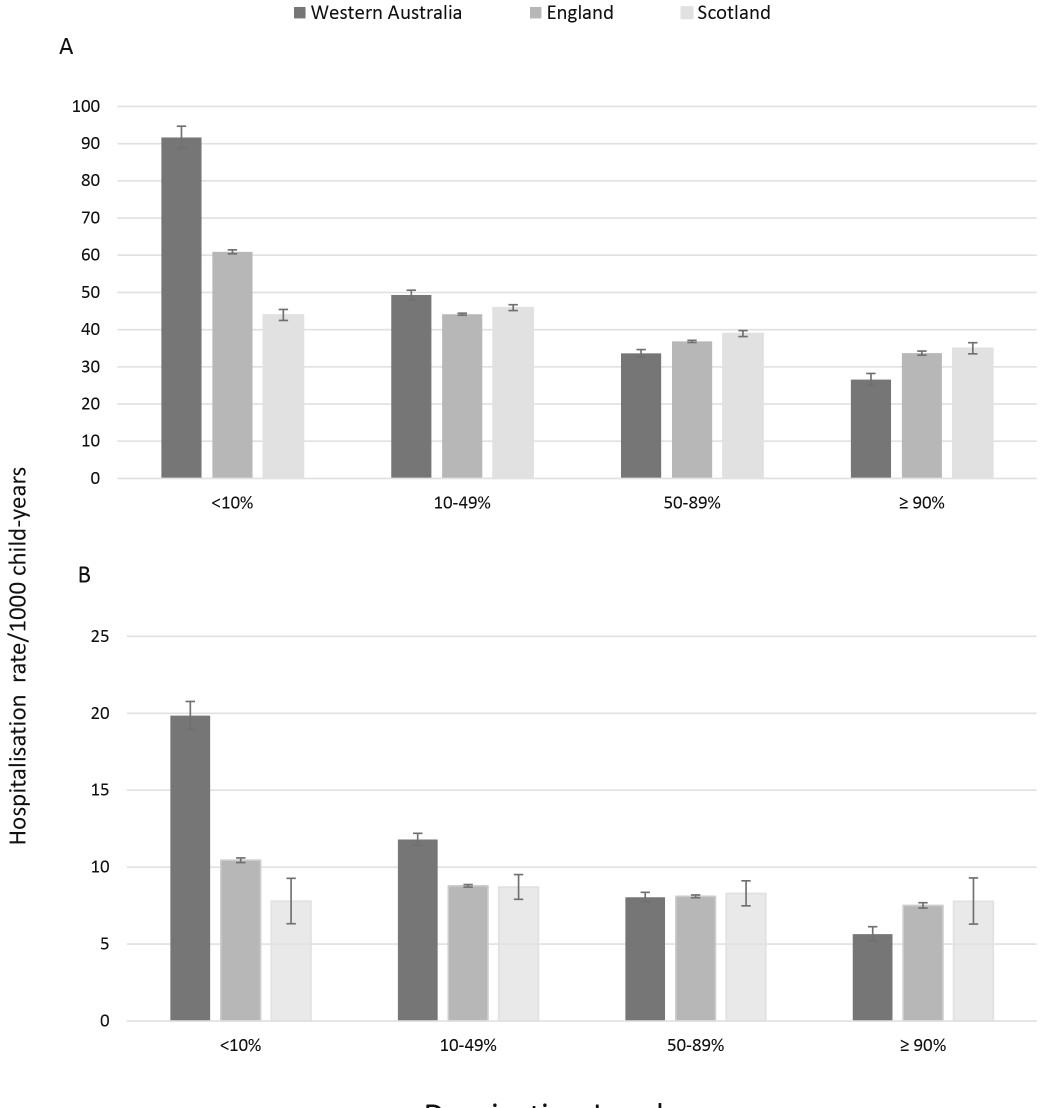

**Figure 1** Hospitalisation rates for ARI in Western Australia, England and Scotland by level of socioeconomic deprivation for (A) infants (<1 year) and (B) young children (1–4 years). Those in the <10% level represent the most deprived and those ≥90% represent those least deprived. ARI, acute respiratory infection.

and more marked in infants compared with children aged 1–4 years (figure 1). The relative difference in ARI hospitalisation rates between the most and least deprived infants was 3.5 (95% CI 3.2 to 3.7) in Western Australia, 1.8 for England and 1.3 for Scotland with similar patterns in children aged 1–4 years (figure 1). In multivariable models, level of socioeconomic deprivation was significantly associated with all ARI categories in all infants but most notably in Western Australia, and in particular, pneumonia (IRR 6.9, 95% CI 5.6 to 8.6) and unspecified ALRI (IRR 8.9, 95% CI 6.7 to 11.8; table 2).

Overall, ARI hospitalisation rates have increased in England and Scotland, but declined (infants) or remained stable (children aged 1–4 years) in Western Australia (figure 2). After adjusting for sex, deprivation and non-ARI emergency hospitalisations, the ARI hospitalisation rate among infants increased by 5% per year in England (IRR 1.05, 95% CI 1.04 to 1.07) and by 3%

per year (IRR 1.03, 95% CI 1.02 to 1.04) in Scotland with no statistically significant trend in Western Australia (IRR 0.99, 95% CI 0.98 to 1.00; table 2, figure 2). Similar results were seen for bronchiolitis admissions in infants.

Diverging trends were seen with pneumonia and unspecified ALRI across the three jurisdictions with pneumonia hospitalisation rates in infants declining in Western Australia from 9.0/1000 in 2002 to 3.9/1000 in 2012 while rates remained steady around 3–4/1000 in England and 2–3/1000 in Scotland (figure 2). After adjusting for sex, socioeconomic deprivation and non-ARI admissions, the annual decline in pneumonia hospitalisations was 6% in Western Australia (IRR 0.94, 95% CI 0.93 to 0.96), 2% in England and 3% in Scotland (table 2). Unspecified ALRI declined in Western Australia annually by 5% but increased by 6% and 2% annually in England and Scotland (table 2).

**Table 2** Risk of hospitalisation for bronchiolitis, pneumonia, unspecified ALRI and overall ARI from log-linear modelling in infants aged <1 year in Western Australia, England and Scotland

| Exposure | Western Australia<br>IRR (95% CI) | England<br>IRR (95% CI) | Scotland<br>IRR (95% CI) |
|---|---|---|---|
| **Bronchiolitis** | | | |
| Year* | 0.99 (0.98 to 1.00) | 1.05 (1.04 to 1.07) | 1.04 (1.03 to 1.05) |
| Male | Reference | Reference | Reference |
| Female | 0.68 (0.64 to 0.72) | 0.68 (0.63 to 0.74) | 0.70 (0.64 to 0.77) |
| Deprivation <10% | 3.34 (3.02 to 3.71) | 1.94 (1.73 to 2.19) | 1.28 (1.16 to 1.42) |
| Deprivation 10%–49% | 2.04 (1.75 to 2.37) | 1.48 (1.07 to 2.06) | 1.29 (0.89 to 1.87) |
| Deprivation 50%–89% | 1.36 (1.19 to 1.55) | 1.18 (0.95 to 1.45) | 1.09 (0.85 to 1.40) |
| Deprivation ≥90% | Reference | Reference | Reference |
| Non-ARI admissions | 1.00 (1.00 to 1.00) | 1.00 (1.00 to 1.00) | 1.00 (1.00 to 1.00) |
| **Pneumonia** | | | |
| Year* | 0.94 (0.93 to 0.96) | 0.98 (0.97 to 0.99) | 0.97 (0.94 to 0.99) |
| Male | Reference | Reference | Reference |
| Female | 0.75 (0.67 to 0.84) | 0.76 (0.70 to 0.82) | 0.80 (0.67 to 0.97) |
| Deprivation 0%–10% | 6.91 (5.59 to 8.56) | 1.47 (1.30 to 1.66) | 1.09 (0.88 to 1.37) |
| Deprivation 10%–49% | 3.26 (2.49 to 4.28) | 0.90 (0.65 to 1.25) | 0.74 (0.39 to 1.43) |
| Deprivation 50%–89% | 1.66 (1.29 to 2.13) | 0.86 (0.70 to 1.07) | 0.80 (0.51 to 1.25) |
| Deprivation >90% | Reference | Reference | Reference |
| Non-ARI admissions | 1.00 (1.00 to 1.00) | 1.00 (1.00 to 1.00) | 1.00 (1.00 to 1.00) |
| **Unspecified ALRI** | | | |
| Year* | 0.95 (0.93 to 0.97) | 1.06 (1.05 to 1.07) | 1.02 (1.00 to 1.04) |
| Male | Reference | Reference | Reference |
| Female | 0.62 (0.54 to 0.71) | 0.65 (0.61 to 068) | 0.73 (0.62 to 0.85) |
| Deprivation <10% | 8.90 (6.69 to 11.83) | 1.81 (1.66 to 1.98) | 0.93 (0.78 to 1.12) |
| Deprivation 10%–49% | 4.18 (2.93 to 5.96) | 1.34 (1.06 to 1.70) | 0.84 (0.48 to 1.48) |
| Deprivation 50%–89% | 1.96 (1.40 to 2.73) | 1.11 (0.95 to 1.30) | 0.85 (0.58 to 1.26) |
| Deprivation >90% | Reference | Reference | Reference |
| Non-ARI admissions | 1.00 (1.00 to 1.00) | 1.00 (1.00 to 1.00) | 1.00 (1.00 to 1.00) |
| **Total ARI** | | | |
| Year* | 0.99 (0.98 to 1.00) | 1.05 (1.04 to 1.06) | 1.03 (1.02 to 1.04) |
| Male | Reference | Reference | Reference |
| Female | 0.68 (0.65 to 0.72) | 0.69 (0.64 to 0.73) | 0.71 (0.65 to 0.77) |
| Deprivation 0%–10% | 3.85 (3.50 to 4.21) | 1.87 (1.70 to 2.06) | 1.25 (1.14 to 1.37) |
| Deprivation 10%–49% | 2.22 (1.95 to 2.54) | 1.41 (1.07 to 1.85) | 1.25 (0.88 to 1.77) |
| Deprivation 50%–89% | 1.42 (1.26 to 1.60) | 1.14 (0.96 to 1.36) | 1.08 (0.85 to 1.36) |
| Deprivation >90% | Reference | Reference | Reference |
| Non-ARI admissions | 1.00 (1.00 to 1.00) | 1.00 (1.00 to 1.00) | 1.00 (1.00 to 1.00) |

*Year included as a linear term.
ALRI, acute lower respiratory infection; ARI, acute respiratory infection; IRR, incidence rate ratio.

## DISCUSSION

ARI, particularly bronchiolitis, continues to be an important cause of infant and childhood hospitalisation. The availability of linked administrative data in three economically similar jurisdictions with publicly funded healthcare systems afforded us the opportunity to compare ARI hospitalisation rates in children. Overall, admission rates were highest in Western Australia and decreasing or remaining stable but increasing in England and Scotland. The relative differences in ARI admission rates between children from the most socioeconomically deprived areas to the least deprived areas were largest in Western Australia.

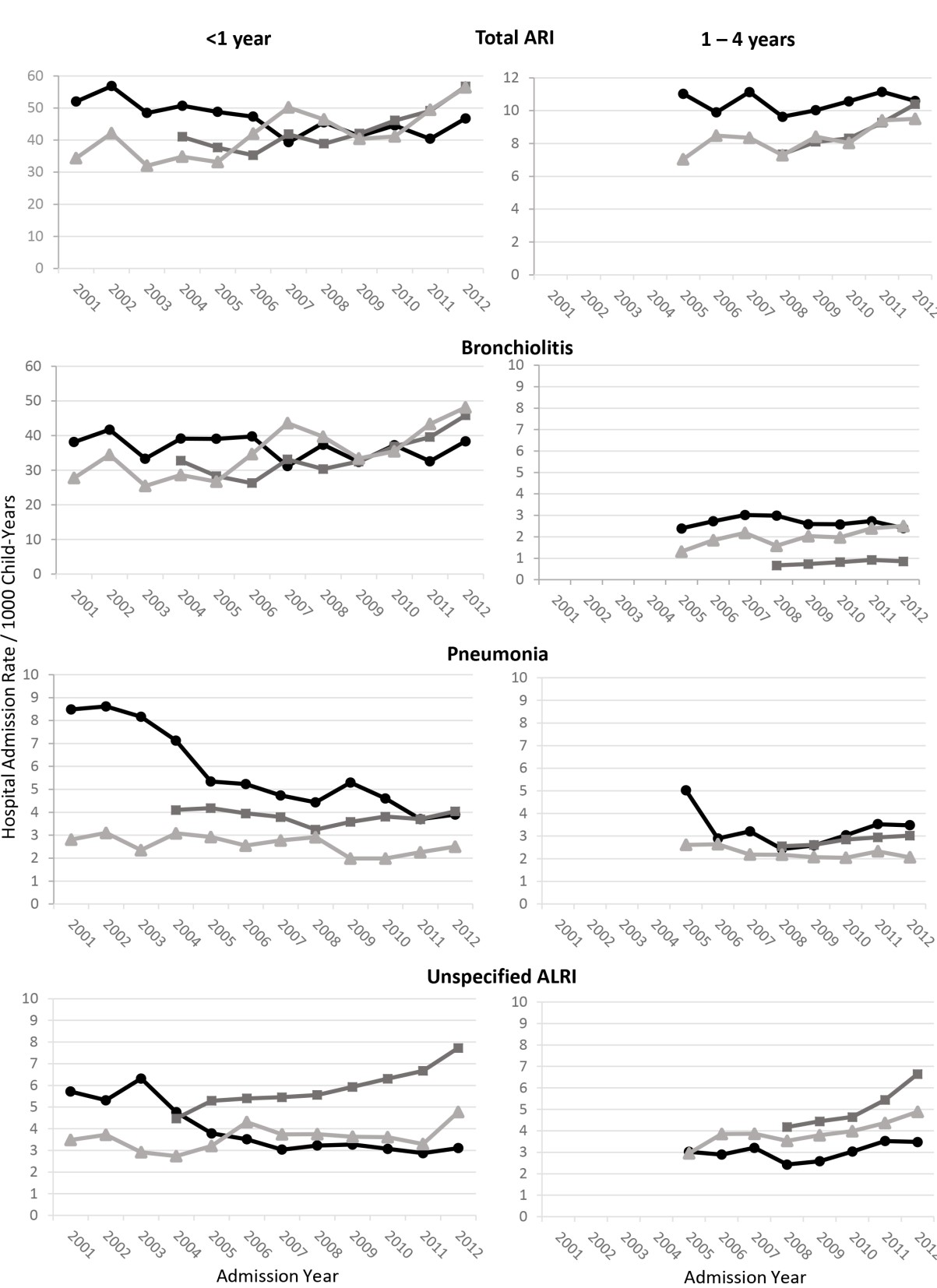

**Figure 2** Hospitalisation rates by year of admission for infants (<1 year) and children (1–4 years) in Western Australia, England and Scotland for ARI, bronchiolitis, pneumonia and unspecified ALRI. ALRI, acute lower respiratory infection; ARI, acute respiratory infection.

The interpretation of hospitalisation trends across countries is complex. We have found higher rates of ARI admissions in Western Australia compared with England and Scotland which could mean a higher incidence in ARI, a higher risk of developing more severe symptoms, or differences in diagnostic coding or hospital admission thresholds. A recent study comparing admission rates between England and Ontario finding substantially higher rates in England was partly explained by differing admission thresholds from differential waiting practices and policies in emergency departments.[4] Comparisons of asthma admissions from national hospital data in Finland and Sweden noted diverging trends citing differences in national coding guidelines and subsequent altered admission thresholds.[31] In an attempt to control for changing admissions thresholds over time within each jurisdiction, we adjusted our multivariable models for the overall trend in non-ARI emergency hospital use. However, we could not adjust for differing thresholds between countries. Emergency hospitalisations are increasing at a faster rate in England compared with other parts of the UK[32] and our data here suggest that hospitalisations due to unspecified ALRI and bronchiolitis in England are contributing to that increase. It is also possible that diverging trends are a result of diagnostic shifts in that for the same clinical presentations, a diagnosis of unspecified ALRI is given in England while other non-specific codes (including codes we have not assessed) are given in Western Australia and Scotland. The use of additional diagnosis codes for ARI seemed more frequent in Western Australia compared with England and Scotland and should be taken into consideration for future comparative studies using ICD diagnosis codes.

Hospitalisation rates for ARI were significantly associated with level of socioeconomic deprivation, consistent with an earlier analysis in England.[33] This association was strongest in Western Australia with IRRs for those in the most deprived level in the order of 3.9 for all ARIs, up to 8.9 for unspecified ALRI. There appeared a linear relationship with level of deprivation and rates of ARI in Western Australia while rates in all levels (bar the most deprived) not differing in England and Scotland. Western Australian data were inclusive of Aboriginal children, an Indigenous population with higher levels of socioeconomic disadvantage[34] compared with their non-Aboriginal peers and a significantly higher burden of pneumonia worldwide,[6 35 36] despite reductions in the 2000s and further reductions seen in our results here, most likely due to the positive impact of pneumococcal vaccination.[6 37] This most likely explains the higher rates of pneumonia seen in Western Australia compared with England and Scotland. We have previously reported that hospitalisation rates for all ARIs are five to seven times higher in young Aboriginal children compared with non-Aboriginal children.[9] Aboriginal children also suffer a disproportionate burden of RSV,[38] the major cause of bronchiolitis which could explain the higher bronchiolitis rates in Western Australia than in England and Scotland.

However, level of socioeconomic deprivation has been associated with hospitalisations for respiratory infections in both Aboriginal and non-Aboriginal children,[9] so the contribution of Aboriginal children alone cannot explain the higher socioeconomic disparities seen here. Indeed, when Aboriginal children were removed from the analysis, the socioeconomic disparities remained, although slightly lessened, and were still higher than England and Scotland (eg, the IRR for most deprived children for all ARI reduced from 3.9 to 2.1 and for unspecified ALRI reduced from 8.9 to 2.9 (data not shown)). Respiratory infections continue to be a source of health inequalities among disadvantaged children worldwide. Geographical remoteness is more of an issue in Western Australia due to its sheer geographical size in comparison to England and Scotland. The lack of adequate primary care in rural and remote Australia[39] which is often coupled with lower socioeconomic levels could be driving higher hospitalisation rates. Nevertheless, these important findings highlight the need for targeted prevention programmes such as smoking cessation, improved housing and timely vaccination for key respiratory pathogens for the most disadvantaged populations in all three jurisdictions.

Since 2013, the UK had been rolling out a universal seasonal influenza vaccination programme for children aged 2–16 years and from 2018, all Australian states and territories offer free seasonal influenza vaccine to children aged between 6 months and 5 years. Our study period was prior to this time. Relative to other ARI diagnoses, recorded influenza hospitalisation rates are low. Assessing the impact of the universal childhood vaccination programme for influenza is likely to be challenging without linking national-level birth cohorts to infection surveillance and vaccination data. This has already been implemented in Scotland.[40] There is also renewed interest in preventing morbidity due to RSV with vaccination.[41] Understanding the baseline hospitalisation rates for RSV–bronchiolitis and pneumonia prior to when vaccination is available is critical to aid in implementation and for its ongoing evaluation postimplementation.

We conducted our analysis on near total population birth cohorts in each jurisdiction and thus our outcome measures have narrow CIs and minimum selection bias. An additional strength of the population-based cohort design is standardisation of analysis protocols and the provision of large numbers allowing us to assess temporal trends and associations with less common infections. The hospital morbidity database systems used in all three jurisdictions have the same population coverage of all inpatient admissions and day surgeries further adding to the validity of our estimates. Although Western Australia is a state within Australia and we have made comparisons to country wide data for England and Scotland, the rate of cross-border hospitalisation from Western Australia to other Australian states is very low.[42]

However, our study does have some limitations. The socioeconomic deprivation scores used were jurisdiction specific and included different items to represent

disadvantage. In addition, area-level socioeconomic deprivation was only measured at birth. Therefore, the observed association between area-level socioeconomic deprivation and the rate of ARI admissions may be subject to increasing measurement error as the child's age increases. How socioeconomic deprivation is associated with morbidity due to ARI at the primary care level is unknown but perhaps likely to aid in explaining disparities in socioeconomic deprivation that we have seen here. While primary care data are more readily available in England and Scotland, limited data with adequate diagnostic information are available for population-based studies in Western Australia. As previously alluded to, there also may be differences in admission thresholds across the three jurisdictions that may explain some higher admission rates across countries. A comparison of emergency department presentations in conjunction with hospitalisations for ARI could be useful here, although diagnostic information from emergency department data is limited,[43] and no individual level data on emergency department visits exist in Scotland. Additionally, through our experience of linking routine laboratory data to hospital data in Western Australia, we are aware of unspecific ICD codes that are associated with detections of respiratory viral pathogens.[44] We did not include such ICD codes (eg, viral infection of unspecified site 'B34') in this analysis. However, we would not expect the potential exclusion of ARI hospitalisations to alter the direction of our results in terms of association with socioeconomic deprivation.

## CONCLUSIONS

Population-based administrative data from economically similar developed countries provide a powerful tool to conduct international comparative studies that can compare and contrast the epidemiology of, and healthcare responses to, respiratory infections. Western Australia experiences higher admissions in children for ARI and a greater disparity in rates according to level of socioeconomic deprivation. Rates are overall slightly lower in England and Scotland but are increasing, particularly in England. These findings suggest that prevention programmes focusing on disadvantaged populations in all three countries are likely to translate into real improvements in the burden of ARI in children. We are planning to use these administrative data to assess effectiveness of interventions (such as vaccination) and how this may affect disparities in ARI admissions rates according to socioeconomic deprivation.

**Author affiliations**
[1]Wesfarmers Centre of Vaccines and Infectious Diseases, Telethon Kids Institute, The University of Western Australia, Perth, Western Australia, Australia
[2]Division of Paediatrics, School of Medicine, The University of Western Australia, Perth, Western Australia, Australia
[3]Department of Infectious Diseases, PrincessMargaret Hospital for Children, Perth, Western Australia, Australia
[4]PathWest Laboratory Medicine WA, QE11 Medical Centre, Perth, Western Australia, Australia
[5]Population, Policy and Practice, University College London Great Ormond Street Institute of Child Health, London, United Kingdom

**Correction notice** This article has been corrected since it was published. The license information has been updated.

**Acknowledgements** We would like to acknowledge the Western Australian Data Custodians, and particularly Alexandra Merchant and Mikhalina Dombrovskaya from the Western Australian Data Linkage Branch, for their assistance and support in collating the data for Western Australia. Source data from England can be accessed by researchers applying to the Health and Social Care Information Centre for England. Copyright © 2017, reused with the permission of NHS Digital. All rights reserved. This research benefits from and contributes to the NIHR Children and Families Policy Research Unit, but was not commissioned by the NIHR Policy Research Programme.

**Contributors** HCM, CCB and PH conceived the study design. PF assisted with data cleaning and coding in Western Australia. AZ and MV assisted with data extraction for England and Scotland. Statistical analysis was conducted by HCM (Western Australia) and PH (England and Scotland) with expert advice from NdK with critical revisions for intellectual content from CCB and RG. HCM drafted the first manuscript with PH. All authors read and approved the final manuscript.

**Funding** This work was supported by a National Health and Medical Research Council Project Grant (1045668), a University of Western Australia Research Collaboration Award (to HCM) and a Wesfarmers Centre of Vaccines and Infectious Diseases Seed Grant (to HCM, CCB, and PH). CCB and HCM are supported by National Health and Medical Research Council Fellowships (1034254 to HCM and 1111596 to CCB). PH was funded by a National Institute for Health Research postdoctoral fellowship, reference number PDF-2013-06-004. AZ's PhD studentship is funded by awards to establish the Farr Institute of Health Informatics Research, London, from the Medical Research Council, Arthritis Research UK, British Heart Foundation, Cancer Research UK, Chief Scientist Office, Economic and Social Research Council, Engineering and Physical Sciences Research Council, National Institute for Health Research, National Institute for Social Care and Health Research, and Wellcome Trust (grant MR/K006584/1). AZ is also supported by the Administrative Data Research Centre for England (funded by the Economic and Social Research Council). MV's PhD studentship is funded by the UBEL DTP of the Economic and Social Research Council. MV was also supported by the Policy Research Unit in the Health of Children, Young People and Families (reference 109/00017), which is funded by the Department of Health Policy Research Programme at UCL. This is an independent report commissioned and funded by the Department of Health.

**Disclaimer** The views expressed in this publication are those of the author(s) and not necessarily those of the NHS, the National Institute for Health Research or the Department of Health.

**Competing interests** None declared.

**Patient consent for publication** Not required.

**Ethics approval** Approval to use the Western Australian data was granted by the Western Australian Department of Health Human Research Ethics Committee, the Western Australian Aboriginal Health Ethics Committee and the Western Australian Data Linkage Branch. We have a data sharing agreement with National Health Service (NHS) Digital to use a deidentified extract of Hospital Episode Statistics for research into children's use of secondary care services; therefore, we did not require ethical approval to use English datasets. For Scotland, approvals were obtained from the Public Benefit and Privacy Panel for Health and Social Care, reference number 1516-0405.

**Provenance and peer review** Not commissioned; externally peer reviewed.

**Data sharing statement** We cannot share the individual-level data used for this study under our agreements with the data providers. The datasets analysed during the current study can be applied for from the Western Australian Data Linkage System (Western Australia; http://www.datalinkage-wa.org.au/), NHS Digital (England; http://content.digital.nhs.uk) and the electronic Data Research and Innovation Service (Scotland; http://www.isdscotland.org/Products-and-Services/eDRIS/). Derived data from these datasets for each jurisdiction are within the paper. No additional data are available.

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
