## [Reviewer comments · BMJ Open]

ARTICLE DETAILS

TITLE (PROVISIONAL)	Temporal trends and socio-economic differences in acute respiratory infection hospitalisations in children: an inter-country comparison of birth cohort studies in Western Australia, England and Scotland
AUTHORS	Moore, Hannah; de Klerk, Nicholas; Blyth, Christopher C.; Gilbert, Ruth; Fathima, Parveen; Zylbersztejn, Ania; Verfürden, Maximiliane; Hardelid, Pia

VERSION 1 - REVIEW

REVIEWER	Bev Lawton Victoria University, Wellington New Zealand
REVIEW RETURNED	22-Jan-2019

GENERAL COMMENTS	The authors have a track record in cohort linkage studies. A statistical review would be appropriate Is there a reason why the population cohorts are not the same years. It would make sense to have them aligned. Was length of hospital stay looked at . It would be useful to know mortality rates for these cohorts. Mortality databases are mentioned but not reported Prematurity is a factor for readmissions as an infant with respiratory infections as well as other complications. Can this be accounted for. Did you look at length of stay? - for example long stay infants are likely to get an infection and hence an ICD code on discharge. Is this accounted for in the analysis. Can you tell if it was an admission or discharge code? Did the cohorts look at minority status- ethnicity , migrant ETC. Methodology Please describe how ethnicity is derived- for example aboriginal as these results are commented on in discussion. Is this a complete or accurate dataset. Australia does not have unique identifier. It would be worth talking about this and the matching done in Western Australia. The Western Australian cohort is quite different from the other cohorts – a state not a country and has a different ethnic make-up , different data collection. How generalisable is it to the rest of Australia? Are admissions tracked if they go elsewhere in Australia. For example born in WA and then move to Melbourne and admitted there. The same could apply when they move in the other cohorts. Was there a rurality index for any of the cohorts? Is health care free for immigrant groups in these cohorts. How accurate was the principal diagnostic code and was there only one diagnosis per discharge. What happens when multiple ICDS of interest are coded for 1 woman .
---

	Why did you break it down into these age ranges- <12months and 1-4 years. Discussion Vaccination comes up frequently There was ne real discussion re this in introduction and in the methodology. Were vaccination rates studied in the cohorts. The overall statement “The study provides insights into the preventable burden of acute respiratory infections” Which codes were defined as preventable and why, timely vaccination data was not collected? Vaccine preventable ICDs ? It is not clear whether being aboriginal or immigrant was associated with higher admission rates. If so discussion re structural determinants of health including racism would be useful.
--	--

REVIEWER	Jill Elaine Torrie Cree Board of Health and Social Services of James Bay, Region 18, Quebec Ministry of Health and and Social Services, Canada
REVIEW RETURNED	27-Jan-2019

GENERAL COMMENTS	I am currently recovering from a concussion so I was just able to read the paper but not able to spend detailed time examining references and the tables as I typically would. So please treat this as a cursory review. Personally, I strongly support this type of administrative-data, non-interventionist research to provide baseline or outcome data in areas amenable to proven preventive action. The authors plan to use the data "to assess effectiveness of interventions (such as vaccination) and how this may affect disparities" is to be encouraged. The link to scales of social deprivation is very pertinent, even with existing methodological caveats, because it provides needed evidence about how to target interventions to act on social determinants. My major concern with the paper is the lack of a summary historical overview of the implementation and reach of vaccination programs for each of the jurisdictions. Are the changes noted related to the implementation of new programs or the poor uptake of existing ones? The authors note that "Vaccination programmes including influenza, pertussis and pneumococcal disease have been implemented in North America, Europe and Australia, but the majority of ARI hospitalisations in high income countries are now caused by non-vaccine preventable viruses including Respiratory Syncytial Virus (RSV), Parainfluenza virus and Human Metapneumovirus.[7]" and as a non-clinician I would have liked to better understand this statement within the context of the data. I wondered about the impact of the Aboriginal children's rates of pneumonia and bronchiolitis on the overall rates in Western Australia. Was it only in the area of economic deprivation where the Aboriginal children's impact was not significant? in other words, the authors suggest that the rates among Aboriginal children are higher but provide no evidence of the non-indigenous Western Australia rates for comparison.
--

VERSION 1 – AUTHOR RESPONSE

Reviewer: 1

1. Is there a reason why the population cohorts are not the same years. It would make sense to have them aligned.

RESPONSE: The cohort for Western Australia and Scotland started earlier due to the availability of data. We were keen to use all available data and therefore did not restrict the years of Western Australia and Scotland to align with the data from England.

2. It would be useful to know mortality rates for these cohorts. Mortality databases are mentioned but not reported.

RESPONSE: Our analyses focused on hospitalisations for acute respiratory infections. Deaths due to acute respiratory infections in these populations are very rare and especially for Western Australia and Scotland, we would not be sufficiently powered to assess mortality rates in this cohort.

3. Prematurity is a factor for readmissions as an infant with respiratory infections as well as other complications. Can this be accounted for. Did you look at length of stay? - for example long stay infants are likely to get an infection and hence an ICD code on discharge. Is this accounted for in the analysis. RESPONSE: We agree that prematurity, along with other perinatal factors, is a risk factor for hospitalisation with acute respiratory infection in children, however prematurity is likely to be on the causal pathway between socio-economic deprivation and acute respiratory infection admission. Perinatal risk factor data were not available for the English and Scottish datasets. It was also not an aim of this paper to assess all risk factors for acute respiratory infections, rather it was primarily to compare population-based hospitalisation rates by acute respiratory infection diagnosis, age, level of socio-economic deprivation and year of admission.

4. Can you tell if it was an admission or discharge code?

RESPONSE: In hospitalisation administration datasets, such as that used in our study across the 3 different jurisdictions, there is not a separate field for an admission or discharge code. As explained on page 7 of the Methods, Outcome Measures section, the hospitalisation dataset in each jurisdiction contained 1 principal diagnosis code which is defined as the principal reason for that hospitalisation, and are coded as such at the time of discharge.

5. Did the cohorts look at minority status- ethnicity, migrant ETC. Please describe how ethnicity is derived- for example aboriginal as these results are commented on in discussion. Is this a complete or accurate dataset. RESPONSE: Due to the different ethnicities and minority groups across the different jurisdictions, this was not assessed in our analyses. Rather the aim of our paper was to assess overall population-based age-specific rates. To aid in the interpretation of the stronger association with socio-economic deprivation and acute respiratory infection hospitalisations seen in Western Australia, analyses were repeated removing Aboriginal and/or Torres Strait Islanders. We have added the following text in the Methods, Data Sources and Study Populations section on page 5-6 on how Aboriginal status was defined in the Western Australian dataset: "Aboriginal status was derived using a validated algorithm using Aboriginal identification information across all available records [16]".

6. Australia does not have unique identifier. It would be worth talking about this and the matching done in Western Australia. The Western Australian cohort is quite different from the other cohorts – a state not a country and has a different ethnic make-up, different data collection. How generalisable is it to the rest of Australia? Are admissions tracked if they go elsewhere in Australia. For example, born in WA and then move to Melbourne and admitted there. The same could apply when they move in the other cohorts.

RESPONSE: The reviewer is correct in that Australia does not have a unique identifier. As stated in the Methods, Data Sources and Study Populations section on page 5, individual-level linkages from administrative datasets is conducted through probabilistic linkage on a set of identifiers. These identifiers include name, date of birth, gender and residential address. This has been standard practice for record linkage research in Australia, and specifically in Western Australia for the past few decades. We have amended this sentence in the methods which now reads: "In the absence of a unique person identifier in Australia, extracted data were probabilistically linked by the Western Australian Data Linkage Branch using a series of demographic identifiers using an established best practice protocol". We have also added an additional reference (Ref 16: Kelman et al Aust N Z J Public Health, 2002) for the linkage practices in Western Australia.

Western Australia covers the western third of Australia and has a similar demographic structure and the same public health system as the rest of Australia. Therefore, the Western Australian results are very likely to be generalisable to the rest of the Australia. The Western Australian data did not capture cross border hospitalisation. As the majority of the Western Australian population resides in the metropolitan region of Perth and on Western coastal regions, cross border hospitalisation is low. We have added a sentence in the discussion at the top of page 14 to reflect this. This now reads: "Although Western Australia is a state within Australia and we have made comparisons to country wide data for England and Scotland, the rate of crossborder hospitalisation from Western Australia to other Australian states is very low.[42]"

7. Was there a rurality index for any of the cohorts? Is health care free for immigrant groups in these cohorts. RESPONSE: As stated at the end of the Introduction on page 5, all jurisdictions have publicly funded health care with free access to primary and public hospital care. Assessing hospitalisation rates by geographical location and remoteness was not an aim of our analysis.

8. How accurate was the principal diagnostic code and was there only one diagnosis per discharge. What happens when multiple ICDS of interest are coded for 1 woman.

RESPONSE: As explained in the Methods, Outcome Measures section on page 6-7, each jurisdiction had 1 principal diagnosis code and up to 20 secondary diagnosis codes in Western Australia, 19 in England and 5 in Scotland. The principal diagnosis code (ie 1 code per hospitalisation) was the diagnosis code that was used when reporting by principal diagnosis code (as in Table 1). Where hospitalisation admission records were combined into episodes of illness (collapsed due to admissions for acute respiratory infections within 14 days of another admission as explained in the Methods, Outcome Measures section on page 7), a hierarchical diagnosis algorithm was developed.

9. Why did you break it down into these age ranges- <12months and 1-4 years.

RESPONSE: It is known that hospitalisation rates for acute respiratory infections are higher in infants aged less than 12 months than those aged older than 12 months. Therefore, it was logical to assess hospitalisations rates in children aged less than 5 years in these 2 age groups.

10. Vaccination comes up frequently. There was ne real discussion re this in introduction and in the methodology. Were vaccination rates studied in the cohorts.

RESPONSE: Individual-level vaccination records were not available in either of the jurisdictions and therefore assessing vaccination rates were not part of the aims of this study and hence not discussed in detail. However, to place the results in context of prevention of acute respiratory infections, we have added in some text at the end of the Introduction, at the top of page 5, to briefly explain the vaccination programs that are available in each jurisdiction specific to respiratory infections. This text is as follows: "Each jurisdiction has established childhood vaccination programs targeting acute respiratory infections. This includes diphtheria, tetanus, pertussis, Haemophilus influenzae type B (3 dose infant schedule), pneumococcal disease (2 + 1 schedule) and recently, seasonal influenza.

Excluding influenza, vaccination coverage at age 12 months is >90% for all 3 jurisdictions.[12, 13]". To further reflect the changing landscape of seasonal influenza vaccination in Australia, an amendment to the sentence "Unlike the United Kingdom, Australia does not have a uniform policy for seasonal influenza vaccination" on page 13 of the Discussion has been made. This now reads: "Since 2013, the United Kingdom has had a universal seasonal influenza vaccination program for children aged 6 months to 9 years and from 2018, all Australian states and territories offer free seasonal influenza vaccine to children aged between 6 months and 5 years. Our study period was prior to this time. Relative to other ARI diagnoses, recorded influenza hospitalisation rates are low. Assessing the impact of the universal childhood vaccination program for influenza is likely to be challenging without linking national-level birth cohorts to infection surveillance and vaccination data."

11. The overall statement "The study provides insights into the preventable burden of acute respiratory infections". Which codes were defined as preventable and why, timely vaccination data was not collected? Vaccine preventable ICDs? It is not clear whether being aboriginal or immigrant was associated with higher admission rates. If so discussion re structural determinants of health including racism would be useful. RESPONSE: The statement in question has been removed from the Article Summary section.

Reviewer: 2

My major concern with the paper is the lack of a summary historical overview of the implementation and reach of vaccination programs for each of the jurisdictions. Are the changes noted related to the implementation of new programs or the poor uptake of existing ones? The authors note that "Vaccination programmes including influenza, pertussis and pneumococcal disease have been implemented in North America, Europe and Australia, but the majority of ARI hospitalisations in high income countries are now caused by non-vaccine preventable viruses including Respiratory Syncytial Virus (RSV), Parainfluenza virus and Human Metapneumovirus.[7]" and as a non-clinician I would have liked to better understand this statement within the context of the data. RESPONSE: As per the response to comment 10 above from Reviewer 1, we have added in some text at the end of the Introduction on page 5 to explain the current vaccination programs in each of the 3 jurisdictions. Further changes have been made in the Discussion on page 13 regarding influenza programs. The beginning of this paragraph now reads: "Since 2013, the United Kingdom has had a universal seasonal influenza vaccination program for children aged 6 months to 9 years and from 2018, all Australian states and territories offer free seasonal influenza vaccine to children aged between 6 months and 5 years. Our study period was prior to this time. Relative to other ARI diagnoses, recorded influenza hospitalisation rates are low. Assessing the impact of the universal childhood vaccination program for influenza is likely to be challenging without linking nationallevel birth cohorts to infection surveillance and vaccination data."

I wondered about the impact of the Aboriginal children's rates of pneumonia and bronchiolitis on the overall rates in Western Australia. Was it only in the area of economic deprivation where the Aboriginal children's impact was not significant? in other words, the authors suggest that the rates among Aboriginal children are higher but provide no evidence of the non-indigenous Western Australia rates for comparison.

RESPONSE: We were not intending to provide detail comparing Aboriginal to non-Aboriginal hospitalisation rates for Western Australia. Rather our focus was to compare total population hospitalisations rates across the 3 jurisdictions in young children. We have added in a sentence and a further reference to the Discussion on page 13 regarding the relative comparisons of acute respiratory infection hospitalisation rates between Aboriginal and non-Aboriginal children from our previous studies in Western Australia. This new sentence reads: "We have previously reported that hospitalisation rates for all acute respiratory infections are 5 to 7 times higher in young Aboriginal children compared with non-Aboriginal children.[9]"